# A comparison of dynamic warm-up and "warm-up" using self-massage tools on subsequent sit-and-reach displacement

**Michele Aquino◉\*⊕, Frederick DiMenna⊕, John Petrizzo‡, George Yusuff‡, Robert M. Otto‡, John Wygand‡**

Department of Health and Sport Sciences, Adelphi University, Garden City, New York, United States of America

⊕ These authors contributed equally to this work.
‡ These authors also contributed equally to this work.
\* maquino@adelphi.edu

## Abstract

### Objectives

A dynamic warm-up (DWU) comprising exercise involving rhythmic muscle actions results in an acute increase in range of motion; however, recent findings suggest that a passive one using self-massage techniques might elicit a similar effect. This study's purpose was to compare the acute effect of leg cycling DWU on sit-and-reach score to the effect of a preparatory regimen of foam rolling (FR) or percussive massage (PM).

### Design

Single-blind, randomized, repeated-measures crossover study.

### Methods

Thirty-two asymptomatic, physically-active participants (male; $n = 17$) aged 20.9 ± 1.5 years performed sit-and-reach tests before, immediately following and 10-, 20- and 30-minutes following eight minutes of each of the "warm-ups." Analyses of variance at each time point across conditions and for the percent change elicited by each intervention were conducted to determine significant differences ($p < 0.05$).

### Results

Repeated-measures ANOVA revealed a significant difference between mean percent difference of sit-and-reach score for FR (8.8 ± 0.5%) compared to DWU and PM ($p = 0.046$ and 0.048, respectively) while DWU (6.3 ± 0.8%) and PM (6.8 ± 0.5%) did not differ ($p = 0.717$). There were no differences between scores across interventions at any of the four time points.

### Conclusions

A bout of FR or PM resulted in an acute increase in a sit-and-reach score during a test performed immediately post and at 10-, 20- and 30-minutes post that was similar in magnitude

**Data Availability Statement:** All relevant data are within the manuscript and its Supporting information files.

**Funding:** The author(s) received no specific funding for this work.

**Competing interests:** The authors have declared that no competing interests exist.

to that which was present following leg cycling. These passive "warm-ups" are appropriate alternate strategies that can be employed to improve performance on a sit-and-reach test.

## Introduction

A DWU comprising exercise that involves rhythmic muscle actions (e.g., cycling or walking/running) has traditionally been used to acutely improve subsequent athletic performance [1]. For example, DWU can facilitate an increase in range of motion by virtue of elevation of core and muscle temperatures that decreases stiffness of the involved muscles and joints [1]. Pre-performance DWU can also provide an ergogenic effect by increasing blood flow/circulation and the response speed of oxidative metabolism (i.e., "$\dot{V}o_2$ kinetics") during high-intensity exercise [2,3]. However, the intensity at which DWU is performed and the time allotted between DWU and the performance bout are important factors to consider regarding whether DWU benefits, impairs or has no effect on performance [1,4]. It also remains to be determined whether "warm-up" strategies other than DWU might be equally or possibly more effective for achieving the desired objective.

Massage has been used to enhance health and provide athletic-performance benefits for thousands of years [5]. Self-massage tools such as foam rolling (FR) and percussive massage (PM) have gained popularity recently. The original interest in these tools was based on their usefulness for managing muscular pain and injury; however, more recently, FR and PM have become popular interventions for reducing the feeling of fatigue and/or delayed-onset muscle soreness after exercise [6]. Consequent to these effects, these and similar acute interventions might be useful as an alternative to DWU for acutely improving subsequent range of motion and athletic performance [7,8].

Studies on whole-body vibration provide some evidence of an ability for the intervention to acutely increase power production; however, the time allotted between application and the performance bout is an important factor to consider [9]. Originally designed for therapeutic purposes, handheld PM treatment has gained popularity in the athletic community in recent years based on the aforementioned positive findings with whole-body vibration [10]. Specifically, PM involves a series of rapid-vibration movements with variations in depth and speed of percussion applied by a handheld machine to a specific area. Such vibratory stimulation has been shown to acutely increase range of motion [8]. For example, when aimed specifically at the hamstrings muscle group, a case study confirmed that PM facilitated reduced hamstrings tightness and improved back range of motion [11]. Although the mechanistic basis(es) for the effect remain(s) to be determined, one possibility is that relaxation reactions occurred due to activation of the Golgi tendon organ [11].

In addition to PM, FR has become popular recently based on its ability to provide benefits similar to those brought about by traditional massage [5,7]. This intervention involves applying pressure on soft tissue using the body's weight pressed onto a myofascial foam roller. Although commonly employed to impact repair and recovery, FR has been increasingly used as a "warm-up" to enhance subsequent performance [12]. Specifically, the friction, shearing forces, fluid exchange and thixotropic effects that are brought on by FR have been reported to restore the muscle length-tension relationship and improve range of motion by breaking down deleterious muscle adhesions [13]. The end result is an improved pliability and range of motion of the targeted musculature [13]. Interestingly, myofascial-release procedures also result in vasodilatory effects with a consequent increase in muscle temperature [13–15].

Collectively, these findings suggest that the acute effects of these interventions are similar to those of DWU. However, there is limited evidence on the effectiveness of tools like these compared to DWU on range of motion during a subsequent testing bout (FR or PM v. DWU) and, to the best of our knowledge, no evidence regarding the effectiveness of these passive techniques compared to the other (FR v. PM).

The purpose of this study was to compare the effects of a traditional DWU performed on a cycle ergometer with those brought about by "warm-ups" using either FR and PM. Our outcome measure was displacement on a sit-and-reach test. We also investigated the time course of any potential effects by conducting multiple sit-and-reach tests for a 30-minute period following each intervention. We hypothesized that both FR and PM would result in similar (with respect to both magnitude and time course) sit-and-reach acute changes in displacement as DWU.

## Methods

### Study design

This single-blind, repeated-measures crossover study, which was conducted in accordance with the ethical principles of the Declaration of Helsinki, was reviewed and approved by the internal review board at Adelphi University. With respect to the COVID-19 pandemic, we followed standard procedures and guidelines as established by the U.S. Government, Adelphi University and Adelphi University's Human Performance Laboratory. The FR and PM procedures were administered unilaterally to the participant's hamstrings, gastrocnemius, gluteus maximus and lumbar extensors for 60 s per side (hence, 120 s per muscle group). Consequently, these "warm-ups" each lasted a total of 8 min. The DWU consisted of leg cycling on an ergometer at 50% of the participant's heart rate reserve (HRR) for 8 min. The three trials were performed on separate days with ≥48 h interspersed and the order of the trails was randomly assigned. Each trial consisted of a baseline sit-and-reach test followed immediately by one of the three interventions. Sit-and reach tests were also conducted at 0, 10, 20 and 30 min following the intervention. This protocol was adapted from one used by Monteiro et al. (2018) [16]. All participants completed a written informed-consent document prior to pre-testing familiarization trials. In order to improve reporting quality within the study, CONSORT updated guidelines for reporting group randomized trials were followed [17].

### Participants

Participant recruitment began on August 14, 2020 and ended on April 13, 2021. An *a priori* sample size calculation ($1-\beta = 0.95$; $\alpha = 0.05$) using G*Power (version 3.1.9.7) found that 18 participants would be required to maintain sufficient power. An "effect size f" of 0.475 was used in this calculation, which was consistent with the effect size reported from the results of Monteiro et al. (2018) [16]. Thirty-two healthy, asymptomatic, physically-active participants (17 male, 15 female) aged 18–30 years (mean ± SD: age, 20.9 ± 1.5 years; height, 169 ± 8 cm; weight, 68.3 ± 9.4 kg; BMI, 23.9 ± 2.9 kg·m$^2$) participated in this study. A physical-activity-readiness questionnaire and a medical-history questionnaire were each completed by all participants prior to testing. Exclusion criteria included: 1.) presence of orthopedic injury and/or fractures; 2.) history of surgery within the past 12 months; 3.) presence of metabolic disorders; and 4.) not aged 18–30 years.

### Study procedures

Prior to completion of the written informed-consent procedure, participants were educated on the potential effects of the interventions. Specifically, we explained that some individuals

can experience minor discomfort 12–48 h following PM [18]. Once written informed consent was obtained and informed consent documents were co-signed by the primary investigator, each participant visited the laboratory so that we could record demographic information and familiarize them with the sit-and-reach test. This preliminary testing session was also used to estimate 50% of their HRR. To do so, we calculated the difference between their estimated maximal heart rate based on their age (i.e., 220 –age) and their resting heart rate, which we measured after having participants lie supine for 10 min. We then calculated 50% of HRR, added that value to the resting HR and recorded that as the heart rate that we would attempt to have participants maintain during their cycling DWU.

Once the preliminary testing session was complete, participants completed the three trial sessions (DWU, FR and PM) in random order. Each trial session began with a baseline sit-and-reach test followed immediately by that session's intervention. Participants then performed the sit-and-reach test at 0-, 10-, 20- and 30-min following completion of the intervention [18]. For all sit-and-reach tests that were performed, the investigator was blinded to the treatment intervention that preceded/followed it.

**Sit-and-reach test.** The sit-and-reach test was conducted utilizing a standard sit-and reach-box. Participants were instructed to sit on the ground in a "long-sit" position while maintaining their torso upright, knees in extension and the plantar surface of their feet in contact with the box. Participants were then told to reach with their arms into a forward-flexed position with the most distal point reached with their fingertips recorded for reach trial. Mean differences from baseline for each intervention pre- and post- were calculated. Percent differences were calculated as follows:

$$\Delta_{\text{POST-PRE}}(\%) = [(\text{POST} - \text{PRE})/\text{PRE}] \text{ x } 100 \tag{1}$$

where POST = the post-intervention sit-and-reach score in inches and PRE = the pre-intervention sit-and-reach score in inches.

For more information about this test being used to assess the acute effect of posterior muscle chain flexibility including photos of it being performed properly, please see Russo et al. [19].

**Foam rolling intervention.** The FR intervention was conducted with participants lying both prone and supine on top of a polyethylene foam roller (12 inches long; 5.5 inches wide) that was placed on the floor. Participants were instructed to use their bodyweight to apply pressure to soft tissue of the hamstrings, gastrocnemius, gluteus maximus and lumbar extensors [18]. The muscles were targeted in the following order: lumbar paraspinals, gluteus maximus, hamstrings, and gastrocnemius. This order was maintained for all participants. Each side of each of those muscle groups was targeted for 60 s resulting in a total "warm-up" duration of 8 min.

**Percussive massage intervention.** The PM intervention was also conducted with participants lying both prone and supine; however, instead of the floor, participants were positioned on a massage bed. A PM tool with a soft, round foam-head attachment was administered to the same musculature as FR (see above) at a frequency of 40 percussions per second. In accordance with recommendations provided by the manufacturer, the investigator began the application proximally and then moved distally before returning to the proximal region. Throughout the intervention, the investigator attempted to apply the same pressure on the skin for each muscle. Once again, each side of each muscle group was targeted for 60 s. The direction and time of application is consistent with the PM manufacturer guidelines [20]. The muscles were targeted as reported for the FR intervention. This order was maintained for all participants.

**Dynamic warm-up intervention.** The DWU was performed on a LODE Excalibur Sport V2.0 cycle ergometer. The participants were instructed to maintain a cadence of 70 – 90 rev·min$^{-1}$ and the ergometer was set in the hyperbolic mode so that work rate remained constant across fluctuations in cadence. The participant's heart rate was measured continuously throughout the cycling bout and an investigator adjusted the load in an attempt to maintain heart rate at approximately ($\pm$5 beats·min$^{-1}$) 50% HRR (see above) once it rose to that level.

## Statistical analysis

Means, standard deviations and all measures of variability were calculated for all participants and performance data. Analysis was performed in order to determine any significant differences between the mean differences from baseline for each intervention. In addition, analysis was performed to determine any significant differences between interventions at each time point (0-, 10-, 20-, 30-min post intervention). Statistical analysis was conducted using repeated-measures ANOVA with a p value $< 0.05$ indicating a significant difference. All statistical analysis was performed using SPSS for Windows Version 28 (IBM Corp, Armonk, NY, USA).

## Results

A flowchart illustrating participant recruitment and progression through the study period is provided in Fig 1 and means $\pm$ SDs for the sit-and-reach displacement scores for the pre- and post-intervention tests are presented in Table 1. A test of assumption was performed to assure adequate utilization of an ANOVA with repeated measures. Mauchly's W, Huynh-Feldt, Greenhouse-Geisser, demonstrated 0.793, 0.870, and .829, respectively [21].

There were no significant differences between scores across interventions at any of the time points (Table 2). However, a repeated-measures ANOVA revealed a significant difference between the mean percent difference from baseline elicited by FR compared to both DWU and PM (p = 0.046 and 0.048, respectively) while DWU and PM did not differ (p = 0.717) (see Figs 2 and 3).

## Discussion

The aim of this study was to compare the effects on sit-and-reach performance brought about by a traditional DWU performed on a cycle ergometer with those facilitated by "warm-ups"

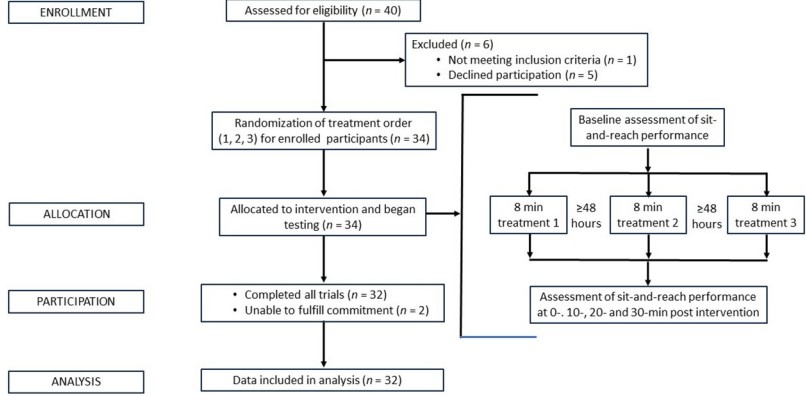

**Fig 1. Participant flow chart.** A flowchart illustrating participant recruitment and progression throughout the study period.

**Table 1. Comparison of sit-and-reach scores prior to, immediately following and at 10-, 20- and 30-minutes following application of the three "warm-up" interventions.**

|                    | PRE          | 0            | 10           | 20           | 30           |
|--------------------|--------------|--------------|--------------|--------------|--------------|
| Cycle Ergometer    | 29.95 ± 7.92 | 31.53 ± 7.33 | 31.93 ± 7.40 | 32.07 ± 7.49 | 31.85 ± 7.57 |
| Foam Rolling       | 29.72 ± 8.17 | 32.52 ± 7.96 | 32.21 ± 8.20 | 32.27 ± 7.91 | 32.40 ± 7.99 |
| Percussive Massage | 30.16 ± 7.83 | 31.98 ± 7.53 | 32.31 ± 7.48 | 32.25 ± 7.38 | 32.35 ± 7.48 |

PRE = the pre-intervention (baseline) sit-and-reach score in inches.

*Data are means ± SD.

using either FR and PM. The primary finding of this study is that compared to a traditional DWU comprising stationary leg cycling, a pre-performance bout involving either FR or PM elicited a similar improvement in subsequent sit-and-reach range-of-motion score that was maintained to the same extent for 30 minutes following the procedure. This confirms our experimental hypothesis and suggests that these self-massage tools can provide an alternative pre-performance preparatory activity if the performance goal is an acute increase in range of motion of the involved joints.

In the present study, we had healthy, young participants complete an eight-minute cycling bout following a test of their sit-and-reach score to determine the effect it would have on subsequent displacement. Consistent with prior research, this traditional DWU elicited an increase; specifically, sit-and-reach score was ~5% greater during the test immediately following the warm-up bout and that increase was maintained for subsequent measurements during the next 30 minutes. Given the nature of this study, we cannot draw conclusions regarding the mechanistic basis(es) responsible for this effect, but it has been suggested that changes like these could be due to cycling-induced arterial dilation which reduces vascular resistance leading to increased blood flow to the musculature involved in the warm-up [1]. A cycling-induced increase in muscle temperature might also decrease the stiffness of muscles and joints thereby allowing for an acute increase in sit-and-reach displacement through the same mechanism [1]. Finally, this type of traditional DWU might decrease muscle stiffness by breaking stable actin and myosin bonds [1]. However, it's important to recognize that an overly intense warm-up of this type with insufficient recovery prior to testing could result in impairments in this and/or other types of performance [1].

In the present study, we found that PM elicited a similar increase in sit-and-reach score as DWU suggesting that at least with respect to this aspect of subsequent performance (i.e.,

**Table 2. Multivariate and within group effects.**

| Multivariate Effects |               | Partial Eta Squared |
|----------------------|---------------|---------------------|
| Intervention         | Wilk's Lambda | 0.196               |
| Time                 | Wilk's Lambda | 0.050               |
| Intervention * Time  | Wilk's Lambda | 0.265               |
| Within Group Effects |               | Partial Eta Squared |
| Intervention         | Linear        | .004                |
|                      | Quadratic     | .194                |
| Time                 | Linear        | .034                |
|                      | Quadratic     | .017                |
| Intervention * Time  | Linear        | .001                |
|                      | Quadratic     | .147                |

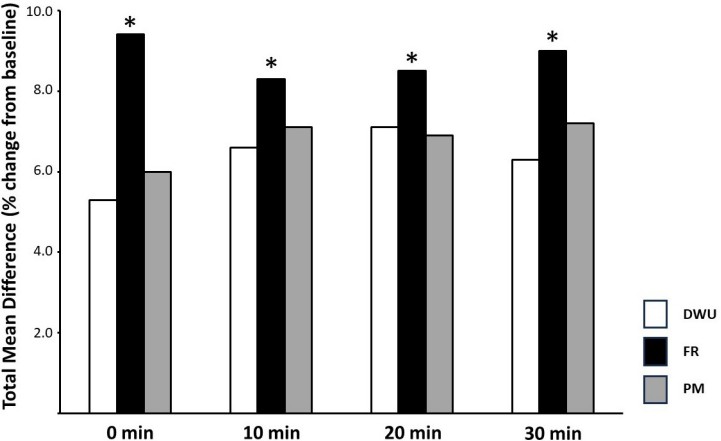

**Fig 2. Mean percent differences.** Mean percent difference from baseline elicited in response to the three preparatory regimens at 0-, 10-, 20- and 30-min post. DWU, dynamic warm-up on a leg-cycle ergometer; FR, self-myofascial release using a foam roller; PM, percussive massage using a handheld device. *P < 0.05 compared with DWU and PM.

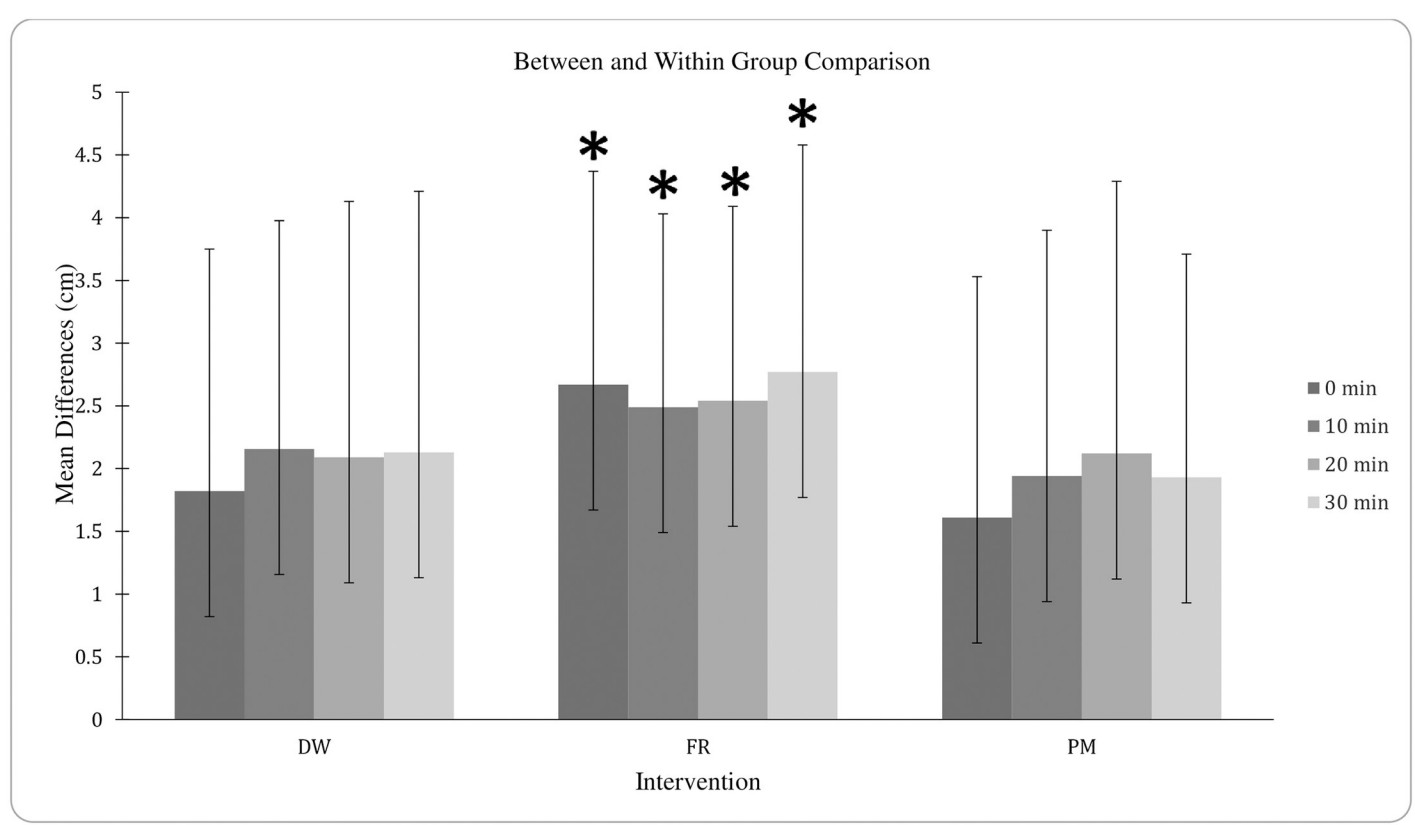

**Fig 3. Gross mean differences between and within groups.** Mean percent difference from baseline elicited in response to the three preparatory regimens at 0-, 10-, 20- and 30-min post. DWU, dynamic warm-up on a leg-cycle ergometer; FR, self-myofascial release using a foam roller; PM, percussive massage using a handheld device. Mean differences were calculated using the data reported in Table 1. *P < 0.05 compared with DWU and PM. *P < 0.05 compared with DWU and PM.

extensibility of the hamstrings and spine), this intervention could be a viable alternative strategy. However, it is likely that compared to DWU, PM exerts its influence via different mechanisms. For example, vibrations that characterize PM could enhance the stretch-reflex loop via activation of muscle spindle primary endings causing reflex inhibition of the antagonist musculature via Ia inhibitory interneurons [22]. Alternatively or in concert, changes induced by PM could be the result of reflex inhibition of the agonist via activation of the Golgi tendon organ [11]. However, in addition to or instead of these changes, it is also possible that PM might simply reduce the perception of stiffness experienced by the recipient immediately after its application [23].

Our findings regarding the influence of PM on subsequent sit-and-reach displacement are in line with previous research. For example, Konrad et al. (2020) showed that a 5-min massage treatment of the calf muscles with a Hypervolt device resulted in an ~18% increase in plantarflexion range of motion compared to no change in the no-intervention test/retest control group [8] while Cochrane & Stannard found that a DWU involving leg cycling and one comprised of whole-body vibration each elicited improvements in sit-and-reach score; however, interestingly, in that study, PM elicited a greater change compared to the DWU (~8% v. ~5%; p < 0.001) [22]. Further research is needed to clarify whether PM with a handheld device like that which was employed in this study elicits a similar effect as whole-body vibration and, if so, whether that modality might indeed be superior to DWU for acutely improving sit-and-reach score.

In addition to PM, we assessed the acute effects of FR on subsequent sit-and-reach score and found that compared to DWU and PM, FR brought a significantly greater mean difference from baseline compared to the other two preparatory procedures (Fig 2). While the clinical relevance of the ~0.8 cm difference we found can be questioned (such a difference is in line with what has been reported for standard error with measurements derived from the test [24]), it is interesting to consider potential differences via which FR might exert a superior influence. Self-myofascial release with FR has been suggested to modulate the activity of the autonomic nervous system by increasing parasympathetic activation and decreasing serum cortisol levels [6,25]. The application of FR might also reduce smooth muscle tension leading to an increase in collagen pliability [24]. Finally, the pressure applied during FR might trigger the release of nitric oxide that brings about a vasodilatory response which improves vascular endothelial function [25]. Regardless of these possibilities, our findings regarding increased sit-and-reach score after FR cohere with those of Smith et al. (2018) who report an increase in sit-and-reach score after FR that was greater than the increase present with the no-intervention test/retest control condition (p = 0.003) [26]. Furthermore, Richman et al. (2019) found that when 5 min of jogging was followed by 6 min of either light walking or FR, FR resulted in an ~4.7% increase (p = 0.002) in sit-and-reach score compared to no change following walking [7]. Interestingly, that finding suggests that the increase that FR facilitated was greater despite the fact that both warm-ups were preceded by jogging (i.e., a DWU). This suggests that FR is indeed superior to DWU for acutely increasing sit-and-reach score so in concert with the (albeit perhaps not of a functionally significant magnitude) greater mean difference we found, the possibility that FR provides for a greater effect should be explored further.

In addition to showing a comparable sit-and-reach score increase for the three different types of preparatory activity immediately following completion, our findings also confirmed that the improvements elicited by FR and PM were maintained to a similar extent compared to those elicited by DWU over the course of the 30-min frame that we measured. These findings extend those that have been reported previously. For example, Dallas et al. (2014) found that applying whole-body vibration prior to sit-and-reach testing brought improvements that lasted for at least 15 min [27] while Smith et al. (2018) found improvements from FR that

lasted for 20 min following cessation of the preparatory activity.[26] In conjunction with what we report, these findings are important for "real-world application" (e.g., in the athletic setting) where FR and PM might not be able to be applied in immediate proximity to the competition location.

Our findings indicating that PM and FR are viable alternatives to DWU have important practical implications. One of the challenging aspects of using an active warm-up like DWU to "prime" subsequent athletic activity is achieving the increase in performance without incurring a decrement due to lingering fatigue from the priming bout. There is a delicate balance between the combination of intensity, duration and proximity to the performance bout that is required to elicit an ergogenic effect and an intensity, duration and proximity which can create an ergolytic one. If reasonably applied, passive warm-up strategies like FR and PM carry no such limitation. However, when interpreting our findings, it is also important to recognize that the term "warm-up" is a general one that is applicable for many different types of athletic preparation. Indeed, other than the common goals of increasing performance and/or decreasing injury risk, there are myriad variables that characterize the different types of performance bouts for which warm-ups are used. Obviously, these must be considered when determining the appropriate preparatory strategy and, specifically, the mechanistic basis(es) of the acute changes that should be targeted. In a literal sense, a warm-up is designed to increase muscle temperature which results in a number of changes in bodily function including an increased rate of enzyme-catalyzed reactions, a rightward shift in the oxyhemoglobin dissociation curve [28], increased ATP utilization [29] and increased nerve-conduction velocity [1]. However, depending upon the specifics of the DWU that is performed, there are other changes that occur independent of or in concert with temperature elevation. For example, a high-intensity DWU of sufficient duration results in lactic acidosis which also facilitates $O_2$ offloading and residual acidosis from DWU has been shown to accelerate oxidative metabolism and performance at higher work rates[3] independent of temperature elevation [30]. Conversely, an intense, but very brief (i.e., not long enough to cause lactic acidosis) high-intensity bout of muscle contraction results in increased force production and rate of force development during subsequent muscle activation (post-activation potentiation) due to an enhanced twitch response that is thought to occur as a result of increased myosin light-chain phosphorylation and/or increased spinal-cord synaptic excitation [31]. With this in mind, including such disparate applications of acute athletic priming under the umbrella term of 'warm-up' is a gross oversimplification; hence, interpretation of our findings should be limited to applications where the specifics of the effect we were looking to elicit (i.e., an acute increase in sit-and-reach displacement) are being targeted.

While we recognize the limited scope of our findings with respect to performance enhancement per se (as opposed to for one specific assessment; i.e., a sit-and-reach challenge), it's also important to consider what prior research reveals about the assessment method we chose as our outcome measure. While it is attractive to speculate that an acute increase in sit-and-reach score indicates enhanced flexibility with the entire posterior chain, it has been shown that the test is valid for measuring hamstrings, but not lumbar extensibility [32]. Accordingly, a "safe interpretation" of our findings is that the acute interventions we employed exerted their effect either predominantly or exclusively on the hamstrings. Our findings, therefore, have relevance for injury prevention in this oft-injured area. For example, Koźlenia and Domaradzki (2021) examined injury occurrence in young athletes (~22 years of age) and found that when their 176-participant cohort was divided into two groups based on functional movement screen test (a test of movement pattern) and sit-and-reach score, the "low flexibility" group (sit-and-reach score $\leq$ 21 cm), were the more frequently-injured group [33]. In another study, these researchers confirmed that this was also the case for physically-active adults. Specifically, sit-and-reach

score alone provides for 41% accuracy in predicting injury risk with the two-group cutoff for these individuals being $\leq$ 15 cm [34]. In addition to recreationally-active individuals, this might also be the case for professional athletes, a group within which hamstring injury and resultant absent days have increased markedly in recent years [35]. Future research can be designed to address the use of acute interventions like the ones we employed for pre-performance warm-up in these types of athletes. It would also be interesting to determine whether these interventions provide a similar influence for both male and female competitors. Generally speaking, females are more flexible than men [36] which means they might possess a smaller window for acute change. This is another important direction for future studies. Finally, being that the mechanism responsible for the changes we observed is a critical factor with respect to interpreting how our findings can be employed in the "real-world" setting, the fact that doing so is beyond the scope of this study indicates that future studies to replicate our findings alongside assessments that allow for potential factors to be tested is another important future direction.

In addition to the caveat regarding applicability of our findings, there are a number of other limitations with the present study that deserve mention. We did not include a control group to account for the effect of repeat sit-and-reach testing per se. However, previous research that included a control group found range-of-motion increases facilitated by warm-up interventions that exceeded what was present in the no-intervention test/retest control condition [27]. Another limitation is the potential for inconsistency within and across participants with respect to the pressure that was applied during FR and PM. For example, for FR, the participant used their bodyweight to apply pressure to soft tissue and despite providing instructions on how to properly do this, it is difficult for an administrator to confirm that they were doing so consistently. For PM, the device we used did not have a pressure setting; hence, we were limited in our ability to precisely control this variable although we did have a single experienced administrator who was instructed to maintain consistency treating all participants. Interestingly, newer versions of the PM device that have the ability to indicate pressure through an analog feedback with bars are now available so it seems logical to suggest that future research exploring this intervention should be conducted with such devices. An important strength of our study was including both FR and PM in our comparison. Indeed, compared to prior investigations which allowed for the effect of either of these interventions to be compared to that elicited by DWU independently, ours also allows for the effects of FR and PM to be compared to each other. This is important because if one of these alternate strategies is superior for eliciting the desired effect, that would be the one which would be preferred.

## Conclusion

The results of this study indicate that a preparatory regimen of either FR or PM elicits an acute increase in range-of-motion score on a sit-and-reach test that is comparable to that which occurs following DWU immediately following and for at least 30 min after its application. Based on these findings, we suggest that FR and PM are viable alternate "warm-up" strategies when the objective is to acutely increase sit-and-reach score. An advantage of these compared to DWU is that when reasonably applied, the fine line between sufficient and excessive intensity, duration and/or proximity to the exercise bout that must be taken into account when prescribing DWU need not be considered. Future research should be designed to identify potential mechanistic drivers of the effects of FR and PM in this regard and, specifically, whether one of these alternate strategies might be superior to the other and/or whether markedly different mechanisms are responsible in which case the effects of the two might be additive.

## Supporting information

**S1 Dataset.**
(XLSX)

**S2 Dataset.**
(XLSX)

## Author Contributions

**Conceptualization:** Michele Aquino, John Petrizzo.

**Data curation:** George Yusuff.

**Formal analysis:** Michele Aquino, Robert M. Otto, John Wygand.

**Investigation:** Frederick DiMenna.

**Methodology:** Michele Aquino, Frederick DiMenna, John Petrizzo, George Yusuff, Robert M. Otto.

**Project administration:** Michele Aquino.

**Software:** Michele Aquino, Robert M. Otto, John Wygand.

**Supervision:** Michele Aquino, Robert M. Otto, John Wygand.

**Writing – original draft:** Michele Aquino, George Yusuff.

**Writing – review & editing:** Michele Aquino, Frederick DiMenna, John Petrizzo.

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
