## [Decision Letter · Decision Letter 0]

22 Apr 2024

PONE-D-24-03963No difference between dynamic warmup and “warmup” using self-massage tools on subsequent sit-and-reach performancePLOS ONE

Dear Dr. Aquino,

Thank you for submitting your manuscript to PLOS ONE. After careful consideration, we feel that it has merit but does not fully meet PLOS ONE’s publication criteria as it currently stands. Therefore, we invite you to submit a revised version of the manuscript that addresses the points raised during the review process.

We look forward to receiving your revised manuscript.

Kind regards,

Holakoo Mohsenifar

Academic Editor

PLOS ONE

Journal Requirements:

Whilst you may use any professional scientific editing service of your choice, PLOS has partnered with both American Journal Experts (AJE) and Editage to provide discounted services to PLOS authors. Both organizations have experience helping authors meet PLOS guidelines and can provide language editing, translation, manuscript formatting, and figure formatting to ensure your manuscript meets our submission guidelines. To take advantage of our partnership with AJE, visit the AJE website (http://aje.com/go/plos) for a 15% discount off AJE services. To take advantage of our partnership with Editage, visit the Editage website (www.editage.com) and enter referral code PLOSEDIT for a 15% discount off Editage services. If the PLOS editorial team finds any language issues in text that either AJE or Editage has edited, the service provider will re-edit the text for free.

Reviewers' comments:

Reviewer's Responses to Questions

**Comments to the Author**

1. Is the manuscript technically sound, and do the data support the conclusions?

Reviewer #1: Partly

Reviewer #2: Yes

2. Has the statistical analysis been performed appropriately and rigorously? 

Reviewer #1: Yes

Reviewer #2: Yes

3. Have the authors made all data underlying the findings in their manuscript fully available?

Reviewer #1: Yes

Reviewer #2: No

4. Is the manuscript presented in an intelligible fashion and written in standard English?

Reviewer #1: Yes

Reviewer #2: Yes

5. Review Comments to the Author

**Reviewer #1:** The manuscript entitled (” No difference between dynamic warmup and “warmup” using self-massage tools on subsequent sit-and-reach performance “) has compared different approaches as warm-up protocols. However, there are some comments as follows.

1. Title: It’s better to change the title as shows comparing methods not the findings of the manuscript.

2. Abstract: Objectives: lines 3, 4: correct the acronym DWU after the word leg cycling. You can use this acronym in the first line of the objectives section.

3. Keywords: In my opinion, “range of motion” is not suitable because indeed, you hadn’t measured ROM and ROM is more related to joints. Also, it’s better to use warm up (or warm-up) instead of warmup especially since you have used the acronym WU not W.

4. In this way, in the abstract conclusion it’s better to say: displacement in sit and reach test.

5. Introduction: You have stated “ We hypothesized that both FR and PM would result in similar (with respect to both magnitude and time course) sit-and-reach acute changes in performance as DWU”, so why you have compared their effectiveness?

6. Please explain what was the rationale for choosing the sit and reach test as the only outcome measure. You targeted different muscle groups including hamstrings, gastrocnemius, gluteus maximus, and lumbar extensors. Indeed this test is more suitable for measuring hamstring flexibility and plantar flexors are not as important as hamstrings.

7. How did you calculate the sample size?

8. In the foam rolling section: add the unit of dimensions.

9. In percussive massage, add the reason for moving from proximal to distal of the limb/area.

10. Mention the order of muscles targeted in FR and PM.

11. Add a section about “the assessment” and explain the details for measuring the score of sit and reach test as your only outcome measure. Also, I recommend adding a figure to show the measuring. How did you convert the score to the percent and so on?

12. Mention the name of the company and country of SPSS software.

13. Did you assess the hamstring length of the participants before being included in the study?

14. Table 1: add the full form of PRE below the table.

15. Please add effect size for each intervention result. It could help to better compare at least within groups.

16. I suggest adding one graph for changes in scores SRT between and within groups. It is more useful than the figure 2. You can present Figure 2 results in a separate table. Also, you can use MANCOVA analysis to omit the baseline difference in the FR group. What is your opinion about this difference?

17. Discussion: It seems that the first group's muscle tissue temperature may be returned to the baseline value before starting the intervention. I suggest presenting your findings also based on different muscle groups at least for FR and PM.

18. Conclusion: you stated “ we suggest that FR and PM are viable alternate “warm-up” strategies when the objective is to acutely increase range of motion. “. Based on my previous suggestions, it should be changed.

**Reviewer #2:** Dear Editor/Authors,

I have reviewed the manuscript titled "No difference between dynamic warmup and “warmup” using self-massage tools on subsequent sit-and-reach performance ". I congratulate the authors for the well-written, organized, and structured manuscript. I thoroughly enjoyed reading it. The Introduction is well-written and structured, addresses all required items related to the research. The Methods section covered every detail regarding participant inclusion/exclusion criteria, measurements, experimental design and protocol, and statistical analysis. The Results sections are well-structured and written. The Discussion is well-written, effectively summarizing the findings and their significance. I suggest adding a paragraph addressing clinical implications and recommendations for future studies. Below are some comments that could further enhance the quality of the manuscript.

Line 25: what does DWU stands for?

Line 39: for what purpose? Please clarify.

Line 85: I recommend using ROM instead of repeatedly mentioning "range of motion" throughout the manuscript to reduce word count.

Line 103: the reference 16 emphasizes on 120s massage. Do you have any reference for 8 minutes?

Line 152. How long did the PM last? Is the same for DWU?

Line 161: Have you checked the assumptions for repeated measure ANOVA. Please report.

Line 190: Please start the first paragraph of your discussion with the aim of the study.

In the discussion section any recommendations for muscle group tested, and mechanisms underlying ROM changes?

Line 284: Can your results be generalized to athletes (professional, etc)? Your sample included male and female. Could one gender benefit more from any of the interventions?

Line 290 – 293: Can you suggest solutions for applying pressure in future studies?

6. PLOS authors have the option to publish the peer review history of their article (what does this mean?). If published, this will include your full peer review and any attached files.

Reviewer #1: No

Reviewer #2: **Yes: **Seyed Hamed Mousavi

---

## [Author Response · Author response to Decision Letter 0]

12 Jun 2024

Dear Academic Editors and Reviewers,

We greatly appreciate your feedback on our work. We have taken the time to thoroughly review, address and respond to your comments. We would like to thank you for your time and efforts. Please find our responses in the red font below. Line numbers have been provided to guide you toward the location of the edits. 

Reviewer #1: The manuscript entitled (” No difference between dynamic warmup and “warmup” using self-massage tools on subsequent sit-and-reach performance “) has compared different approaches as warm-up protocols. However, there are some comments as follows.

We would like to thank Reviewer #1 for providing comments. We have addressed each and believe that the manuscript is improved as a result. We look forward to more feedback and, specifically, Reviewer #1’s views regarding whether we have addressed his/her concerns adequately.

1. Title: It’s better to change the title as shows comparing methods not the findings of the manuscript.

Thank you for this suggestion. We have changed the title of the manuscript to: “A comparison of dynamic warm-up and “warm-up” using self-massage tools on subsequent sit-and-reach displacement.”

2. Abstract: Objectives: lines 3, 4: correct the acronym DWU after the word leg cycling. You can use this acronym in the first line of the objectives section.

We have now established the acronym for “dynamic warm-up” when it is first mentioned in the Objectives section and then referred to it as “DWU” from that point forward. 

3. Keywords: In my opinion, “range of motion” is not suitable because indeed, you hadn’t measured ROM and ROM is more related to joints. 

We have changed “ROM” to “sit-and-reach displacement” or “sit-and-reach score” in the Keywords and throughout the rest of the manuscript.

Also, it’s better to use warm up (or warm-up) instead of warmup especially since you have used the acronym WU not W.

Good point. We have changed “warmup” to “warm-up” in the title and throughout the manuscript.

4. In this way, in the abstract conclusion it’s better to say: displacement in sit and reach test.

Thank you. We have already addressed this in response to your comment above.

5. Introduction: You have stated “ We hypothesized that both FR and PM would result in similar (with respect to both magnitude and time course) sit-and-reach acute changes in performance as DWU”, so why you have compared their effectiveness?

Our hypothesis was formulated based on previous findings; however, as we point out in the Discussion, our methodology allowed us to confirm, clarify and extend what had been shown previously. For example, no previous study compared all three so while it seemed reasonable to combine results from previous studies to make research-based speculation regarding what we would find, we believe it was important to test hypotheses formulated accordingly to confirm that the reasonable speculation was indeed factual.

6. Please explain what was the rationale for choosing the sit and reach test as the only outcome measure. You targeted different muscle groups including hamstrings, gastrocnemius, gluteus maximus, and lumbar extensors. Indeed this test is more suitable for measuring hamstring flexibility and plantar flexors are not as important as hamstrings.

Thank you for providing this suggestion. We have added an additional paragraph to the Discussion with references to previous research which we believe supports our choice of the sit-and-reach test as a robust (albeit singular) outcome measure. Please see lines 445-476 in the revised version of the manuscript.

7. How did you calculate the sample size?

We want to thank the reviewer for this question. We have provided our a priori power analysis, please find this addition from lines 125-129.

8. In the foam rolling section: add the unit of dimensions.

We actually have the units of dimension (inches) stated; however, we did so by using an abbreviation that might have created confusion (in). In the revised version of the manuscript, we have written out :inches” to avoid any such confusion. 

9. In percussive massage, add the reason for moving from proximal to distal of the limb/area.

Thank you for this suggestion. We have provided the rationale for moving from proximal to distal. This inclusion can be found on lines 171-172.

10. Mention the order of muscles targeted in FR and PM.

Thank you. The order of the muscles targeted in FR and PM were included. These additions can be found on lines 160 - 162 and 174 – 175. 

11. Add a section about “the assessment” and explain the details for measuring the score of sit and reach test as your only outcome measure. Also, I recommend adding a figure to show the measuring. How did you convert the score to the percent and so on?

We have added the information to the Methods section (please see lines 157-171) along with a reference with more information about this test including photos of it being performed properly.

Equation 1: ∆POST-PRE (%) = [(POST – PRE)/PRE] x 100 where:

POST = the post-intervention sit-and-reach score in inches

PRE = the pre-intervention sit-and-reach score in inches

12. Mention the name of the company and country of SPSS software.

We have added this information to the revised version of the manuscript.

13. Did you assess the hamstring length of the participants before being included in the study?

Thank you for this query. Hamstrings length of the participants before being included in the study was not conducted. All assessments were compared to the participants respective baseline for each intervention; therefore a preliminary “inclusion” hamstring length was not performed. 

14. Table 1: add the full form of PRE below the table.

We have defined “PRE” in the revised version of the manuscript as per your request.

15. Please add effect size for each intervention result. It could help to better compare at least within groups.

Thank you. We have provided in Table 2, the Multivariate and Within Group effects for Intervention, Time, and the Time x Intervention interaction. 

16. I suggest adding one graph for changes in scores SRT between and within groups. It is more useful than the figure 2. You can present Figure 2 results in a separate table. Also, you can use MANCOVA analysis to omit the baseline difference in the FR group. What is your opinion about this difference?

Thank you for this suggestion. Figure 3 has now been added to demonstrate gross mean differences between and within groups. In the results section we have reported tests of sphericity, to assure ANOVA test of assumptions for repeated measures were adequate. Therefore, given the repeated measure nature of the design and the results of the tests of sphericity, we do not believe a MANCOVA would have been required for this data set. 

17. Discussion: It seems that the first group's muscle tissue temperature may be returned to the baseline value before starting the intervention. I suggest presenting your findings also based on different muscle groups at least for FR and PM.

We apologize, but we do not understand your suggestion. The post sit-and-reach assessments were performed at 0, 10, 20 and 30 min following the intervention so we believe that muscle temperature would have been elevated for our DWU condition for at least the initial post assessment. As for FR and PM, as we point out, our methodology does not allow for us to draw conclusions about the mechanistic basis(es) that underpin(s) the effect we observed; hence, whether or not temperature was elevated for any of the post assessments cannot be determined. If this is not what you were referring to, please provide more information regarding changes and/or additions you would suggest that we make to the manuscript to address the concern you have explained above. 

18. Conclusion: you stated “ we suggest that FR and PM are viable alternate “warm-up” strategies when the objective is to acutely increase range of motion. “. Based on my previous suggestions, it should be changed.

As previously mentioned, we have removed reference to range of motion and replaced it with “sit-and-reach displacement” or “sit-and-reach score” throughout the revised version of the manuscript including in the Discussion.

Reviewer #2: Dear Editor/Authors,

I have reviewed the manuscript titled "No difference between dynamic warmup and “warmup” using self-massage tools on subsequent sit-and-reach performance ". I congratulate the authors for the well-written, organized, and structured manuscript. I thoroughly enjoyed reading it. The Introduction is well-written and structured, addresses all required items related to the research. The Methods section covered every detail regarding participant inclusion/exclusion criteria, measurements, experimental design and protocol, and statistical analysis. The Results sections are well-structured and written. The Discussion is well-written, effectively summarizing the findings and their significance. 

We would like to thank Reviewer #2 for taking the time to read our manuscript and provide helpful feedback. We are glad to learn that he/she enjoyed reading our manuscript and appreciated our efforts. We have addressed each concern and look forward to more (hopefully positive) feedback.

I suggest adding a paragraph addressing clinical implications and recommendations for future studies. Below are some comments that could further enhance the quality of the manuscript.

Thank you for providing this suggestion. Interestingly, it is similar to one that Reviewer 1 made so to address each of your concerns, we have added an additional paragraph to the Discussion with references to previous research which we believe supports our choice of the sit-and-reach test as a robust (albeit singular) outcome measure. Please see lines 445-476 in the revised version of the manuscript.

Line 25: what does DWU stands for?

In response to a concern of Reviewer #1, we have now defined “DWU” in the first sentence of the Objective section.

Line 39: for what purpose? Please clarify.

We have clarified our purpose in the revised version of the manuscript (“These passive “warm-ups” are appropriate alternate strategies that can be employed for this to improve performance on a sit-and-reach test.

Line 85: I recommend using ROM instead of repeatedly mentioning "range of motion" throughout the manuscript to reduce word count.

We have removed reference to “range of motion” in response to Reviewer #1’s concern and now refer to sit-and-reach displacement here and throughout.

Line 103: the reference 16 emphasizes on 120s massage. Do you have any reference for 8 minutes?

Thank you. The 8 minutes of total intervention fell within the time frame of the PM manufacture guidelines, which ultimately was the 120s per muscle group (Lumbar Extensors, Gluteus Maximus, Hamstrings, Gastrocnemius). We have referenced the manufactures guidelines in lines 186-187, under the Percussive Massage Heading. 

Line 152. How long did the PM last? Is the same for DWU?

The PM intervention lasted a total of 8 minutes, which was the same for DWU and FR. 

Line 161: Have you checked the assumptions for repeated measure ANOVA. Please report.

In the results section we have reported tests of sphericity, to assure ANOVA test of assumptions for repeated measures were adequate. 

Line 190: Please start the first paragraph of your discussion with the aim of the study.

We have added a sentence with the aim of the study as the first sentence of the Discussion as you have requested.

In the discussion section any recommendations for muscle group tested, and mechanisms underlying ROM changes?

Thank you for these questions. In the additional paragraph we have added to the Discussion, we have provided an explanation regarding the importance of the muscle groups we tested with regard to athletic performance per se (as opposed to simply achieving a higher score on a sat-and-reach assessment. As for potential mechanisms underpinning the changes in sit-and-reach performance that we did observe, unfortunately, our methodology does not allow us to draw any definitive conclusions. We do, however, offer some speculation based on previous findings (for example, see Discussion lines 413-444) along with a mention regarding future research designed to provide more insight.

Line 284: Can your results be generalized to athletes (professional, etc.)? Your sample included male and female. Could one gender benefit more from any of the interventions?

More good questions! And more material that we have covered in the newly-added penultimate paragraph of the Discussion.

Line 290 – 293: Can you suggest solutions for applying pressure in future studies?

In addition to our recommendations about future research to explore the mechanistic basis(es) that underpin(s) the increase in sit-and-reach score that we observed following both FR and PM, we have made mention of the fact that newer versions of the PM device that can gauge pressure through an analog feedback with bars are now available (see lines 493-496 in Discussion). Moderating PM in accordance with this feedback might provide important feedback for figuring out what does indeed drive its influence.

---

## [Editor Report · Decision Letter 1]

1 Jul 2024

A comparison of dynamic warm-up and “warm-up” using self-massage tools on subsequent sit-and-reach displacement

PONE-D-24-03963R1

Dear Dr. Michele Aquino, 

We’re pleased to inform you that your manuscript has been judged scientifically suitable for publication and will be formally accepted for publication once it meets all outstanding technical requirements.

Kind regards,

Holakoo Mohsenifar

Academic Editor

PLOS ONE
---

## [Editor Report · Acceptance letter]

7 Jul 2024

PONE-D-24-03963R1 

PLOS ONE

Dear Dr. Aquino, 

I'm pleased to inform you that your manuscript has been deemed suitable for publication in PLOS ONE. Congratulations! Your manuscript is now being handed over to our production team.

Kind regards, 

on behalf of

Dr. Holakoo Mohsenifar 

Academic Editor

PLOS ONE